# The Roles of YAP/TAZ and the Hippo Pathway in Healthy and Diseased Skin

**DOI:** 10.3390/cells8050411

**Published:** 2019-05-03

**Authors:** Emanuel Rognoni, Gernot Walko

**Affiliations:** 1Centre for Endocrinology, William Harvey Research Institute, Barts and The London School of Medicine, Queen Mary University of London, London EC1M 6BQ, UK; e.rognoni@qmul.ac.uk; 2Department of Biology and Biochemistry & Centre for Therapeutic Innovation, University of Bath, Claverton Down, Bath BA2 7AY, UK

**Keywords:** Hippo signalling, skin development, stem cells, skin cancer, fibroblasts, fibrosis, wound healing

## Abstract

Skin is the largest organ of the human body. Its architecture and physiological functions depend on diverse populations of epidermal cells and dermal fibroblasts. Reciprocal communication between the epidermis and dermis plays a key role in skin development, homeostasis and repair. While several stem cell populations have been identified in the epidermis with distinct locations and functions, there is additional heterogeneity within the mesenchymal cells of the dermis. Here, we discuss the current knowledge of how the Hippo pathway and its downstream effectors Yes-associated protein (YAP) and transcriptional coactivator with PDZ-binding motif (TAZ) contribute to the maintenance, activation and coordination of the epidermal and dermal cell populations during development, homeostasis, wound healing and cancer.

## 1. Introduction

The skin is the largest organ of the human body. It forms a protective interface between the body and the external environment. Various skin cell populations act in harmony to provide protection from daily wear and tear, harmful microbes and other assaults from the external environment. In addition, skin forms a barrier against water loss, enables thermoregulation, and relays somatosensory information to the brain to inform reflexes and behaviours [1]. 

The outermost tissue layer of the skin, the epidermis, is a multilayered (stratified) squamous epithelium that is separated by a basement membrane (BM) from the underlying dermis. The dermis forms the skin scaffold consisting of a dense extracellular matrix (ECM) meshwork and different cell populations, including fibroblasts, sensory neurons and endothelial and immune cells. Maintenance and repair of the epidermis are dependent on stem cells (SCs) that both self-renew and give rise to cells that undergo terminal differentiation. Multiple SC populations have been identified within mammalian epidermis; these are distinguished by their location and the markers that they express [2,3,4,5]. However, during tissue regeneration such as wound healing, these SC populations display remarkable plasticity by switching from one SC type to another or by generating a wider range of differentiated lineages than under steady state conditions [1,6]. Likewise, an astonishing heterogeneity and plasticity of dermal cell populations has recently been identified in mouse and human dermis during development and tissue regeneration, which seems to evolve with age [7,8,9,10,11]. Upon tissue damage dermal fibroblasts not only become activated, start proliferating and deposit/ remodel ECM, but are also involved in regenerating HF and the hypodermis [3]. Plasticity in the epidermis and dermis typically resolves as wounds heal. However, in cancer, it can endure [1,6].

Carcinomas originating from the epidermis are by far the most frequently diagnosed human cancers, with cutaneous squamous cell carcinoma being the most dangerous subtype due to its ability to metastasise [12,13]. There is now strong evidence that epidermal cancers are maintained by a subpopulation of epidermal cells termed tumour-initiating cells or cancer SCs, which hijack the homeostatic controls operating in normal epidermal stem cells to endow themselves with the potential to self-renew and fuel the cancer [14,15]. Distinct epithelial SC populations have been associated with distinct tumour types; however, how different fibroblast lineages affect tumour development, type and progression is still unclear [3]. 

In the epidermis and dermis, the Hippo signalling pathway and its downstream effectors, the transcriptional coactivators Yes-associated protein (YAP) and transcriptional coactivator with PDZ-binding motif (TAZ, also called WW Domain Containing Transcription Regulator 1 (WWTR1)), regulate diverse tissue-specific functions during development, homeostasis and regeneration. While in the epidermis YAP/TAZ are essential to control cell growth and differentiation, in fibroblasts they function predominantly as an intracellular mechanical rheostat to sense the physical cellular environment, promoting ECM remodelling and contractility. The Hippo pathway is a tumour suppressor pathway, since its dysregulation and the resulting YAP/TAZ hyperactivation promote cancer development.

This review is divided into three parts. We first give an overview of the contributions of the different epidermal SC and dermal fibroblast populations to skin tissue development, homeostasis and repair, and the roles of these cells in cancer development (chapters 2–6); in the second part, we review the existing work on the specific roles of YAP/TAZ in controlling epidermal SC and fibroblast functions in healthy and diseased skin, summarise the various mechanisms of YAP/TAZ regulation and discuss YAP/TAZ-targeting therapy-approaches (chapters 7–11). We finally close this review with some future perspectives for YAP/TAZ in skin biology (chapter 12).

## 2. Skin Architecture

The skin consists of two tissue layers—the epidermis and the underlying dermis—which are separated by a BM (Figure 1). The epidermis is a multilayered (stratified) squamous epithelium consisting of the IFE and associated hair follicles (HFs), sebaceous glands (SGs), and sweat glands. Keratinocytes are the main epidermal cell type. Several other cell types, such as Merkel cells, melanocytes, gamma delta (γ∂) T-cells and Langerhans cells, are also found in mammalian epidermis [1,3,5]. 

The interfollicular epidermis (IFE) consists of an inner layer of proliferative basal cells that express keratins Krt5 and Krt14 and are attached to the underlying BM via integrin (ITG) extracellular matrix (ECM) receptors and several layers of suprabasal keratinocytes at various stages of a terminal differentiation programme. Only the innermost (basal) layer is proliferative (Figure 2A). To constantly renew the epidermal barrier, differentiating cells moving outward replenish the terminally differentiated squames that are sloughed from the skin surface [1,5]. 

Contiguous with the IFE are HFs (Figure 1), which are encased by the BM and frequently cycle between growth and resting phase. Within the HF epithelium multiple SC populations have been identified through the use of lineage tracing and flow cytometry (for a comprehensive review on lineage tracing see [16,17]). These include SCs of the junctional zone between the IFE, HF and SG (Figure 1), which express the receptor tyrosine kinase regulator Lrig1, and cells of the lower hair follicle that express the R-spondin receptor Lgr5, the transmembrane phosphoglycoprotein CD34, keratins Krt15 and Krt19 and Sox9 [2,18]. In addition, Gli1^+^ and Lgr6^+^ stem cells are found in the upper hair follicle and with the latter scattered within the IFE [2,18]. 

Upon growth induction, cells in the HF lower bulge become activated, proliferate and differentiate in the district hair layers through a complex mesenchymal–epithelial cross-talk, while the lower HF portion expands [3,5]. The upper portion of the HF, which includes the infundibulum, junctional zone and sebaceous glands, does not cycle but undergoes frequent turnover governed by multiple resident SC pools, each responsible for maintaining homeostasis of their nearby territory [2,3]. Similar to the IFE, in the SG proliferating cells anchored to the BM support turnover of differentiated cells [2]. In contrast to mouse skin, the epidermis in human skin is proportionately much more interfollicular, is much thicker and the epidermis–dermis junction is undulated [19]. These epidermal invaginations are referred to as rete ridges and their width and depth varies with age and disease [20]. Human IFE SCs have been shown to reside in cluster between rete ridges and decline with age [20,21,22,23]. 

The dermis is composed of different sublayers that are distinguished by cell type, cell density, and extracellular matrix (ECM) composition (Figure 1) [3,24]. The papillary dermis is located closest to the IFE and displays a high fibroblast density. The reticular dermis is the central and largest layer of the dermis, and consists of a thick, highly organised collagen fibre-rich ECM with sparsely embedded fibroblasts. Under the reticular dermis lies the dermal white adipose tissue (DWAT), also referred to as hypodermis, which harbours pre- and mature adipocytes [3]. In addition specialised fibroblast subpopulations are associated with the blood vessels (pericytes) and HFs giving rise to the dermal sheath, dermal papilla and the arrector pili muscle (APM) (Figure 1). The dermis is highly vascularised and innervated, and cells of the immune system traffic through both the dermis and epidermis [24].

## 3. Skin Morphogenesis and Hair Follicle Development

Shortly after gastrulation, the skin forms as a flat single-layered epithelium from the surface ectoderm and the dermis (arising from the mesoderm) appears homogenous in composition [5]. Initially, unspecified epidermal progenitors divide exclusively parallel to the BM underneath, but within several days, divisions become first oblique and then more perpendicular, leading to asymmetric fates, stratification, and differentiation of the epidermis [5] (Figure 2B). Local induction of WNT signalling in the epidermis and subsequently in the dermis causes epidermal cells with high levels of WNT signalling to cluster into HF placodes, which are characterised by expression of adult SC markers such as the transcription factor (TF) Sox9, and Lrig1 and Lgr6 [5,16]. Production of sonic hedgehog (SHH) by WNT^hi^ cells then induces further HF maturation, during which SC markers begin to segregate into the distinct HF compartments including the HF bulge, junctional zone, isthmus and sebaceous gland [2,18] (Figure 2B). By the time epidermal morphogenesis is complete, the SCs of the different HF compartments then reside in discrete niches whose cellular components and other sources of signalling factors heavily influence their behaviour [1,3,5].

During embryonic development multipotent fibroblasts are highly proliferative and start to differentiate into papillary and reticular lineages [7,9]. While the papillary fibroblasts give rise to the dermal sheath, dermal papilla and APM, reticular cells differentiate to preadipocyte and mature adipocytes [7,9]. Postnatally, fibroblasts stop proliferating and enter a quiescent state for efficient ECM deposition and remodelling. During dermal maturation fibroblast lineages become segregated by increased ECM deposition, start intermixing and lineage marker expression dynamically changes with age. While dermal fibroblast density and the papillary layer decrease with age, there is an increase in adipocyte layer [11,25,26]. Intriguingly, it was shown that the coordinated switch in fibroblast behaviour from being highly proliferative in embryonic development to quiescence postnatally in order to allow efficient ECM deposition/remodelling is balanced by a negative feedback loop which is necessary and sufficient to define dermal architecture during development [27]. So far, the cell intrinsic and extrinsic regulatory mechanisms controlling dermal fibroblast lineage identity, behaviour and fate, are largely unknown. 

## 4. Skin Homeostasis and Tissue Repair

Homeostasis of the IFE is maintained by a delicate balance between basal cell proliferation and suprabasal cell differentiation/stratification (Figure 2A). At the onset of differentiation, basal cells become detached (delaminate) from the BM, stop proliferating, and once located in the suprabasal cell layer, start executing terminal cell differentiation programmes that involve extensive remodelling of intracellular proteins, intercellular junctions, lipid extrusion and nuclear fragmentation. These events culminate in the terminally differentiated cells becoming highly cross-linked scales that are exfoliated from the surface of the skin [28,29]. Intriguingly the temporal and spatial dynamics of differentiation commitment seems to be controlled by autoregulatory network of phosphatases [30].

While several SC populations have been described in the HFs which maintain homeostasis and are able to participate to wound repair, the identity and organisation of SCs within the IFE is still an unresolved matter. Early studies of human and mouse epidermis revealed heterogeneity in the propensity of basal IFE cells to proliferate, and the concept arose that IFE SCs self-renew infrequently, while their progeny undergo a small number of amplifying divisions prior to the onset of terminal differentiation [31,32]. Such so-called transit amplifying cells were also identified in vitro in studies of colony formation by cultured human epidermal cells [33,34]. While it was first believed that the IFE is maintained by a single cell population [35] further studies showed that basal cells in mouse epidermis are heterogeneous and some exhibit SC characteristics [36,37]. Single-cell transcriptomic analysis of cultured human epidermal cells [38] and mouse and human epidermis [39,40] further supports the concept of IFE SC heterogeneity. 

In the dermis, clonal lineage tracing revealed that fibroblast turnover is very low lineages and lineages remain in a quiescence state long-term to deposit and remodel their surrounding ECM during homeostasis. If mesenchymal SC actively contribute to dermal homeostasis is still a matter of debate [41]. 

During wound healing different cell types of the epidermis and dermis are coordinated to regenerate the skin [3,4]. In the initial inflammatory phase, a blood clot forms and immune cells infiltrate the wound bed, which is followed by epidermal and dermal cells starting to proliferate and to migrate into the wound bed in the proliferative phase. During resolution phase, epidermal cells still proliferate, while dermal fibroblasts enter quiescence to allow for efficient ECM deposition and remodelling. Recently it was shown that epidermal cells organise in concentric zones with distinct cellular activity and gene expression [42,43]. In the wound bed central migratory zone, cell migration and differentiation are tightly coordinated to promote epidermal thickening and wound closure. In the second (proliferative) zone cells are highly proliferative, polarised towards the migration direction and control the involvement of surrounding epithelial cells in the unwounded area. Although different epidermal cells from the IFE and HF contribute to wound repair, within the wound healing zones they exhibit similar behaviour in cell proliferation, migration and differentiation. Indeed, during wound healing, SCs exhibit a high degree of plasticity and temporarily lose their lineage restriction [36]. In addition, it was recently shown that differentiated sebaceous duct cells are able to dedifferentiate, proliferate and regenerate the IFE long-term, however the transcriptional and epigenetic mechanisms are still unclear [44].

Upon wounding different fibroblast lineages become activated (α-smooth muscle actin (aSMA)-positive myofibroblasts) at the wound site, quickly resume proliferation and migrate into the wound bed [11,27]. While fibroblast in the reticular and adipocyte layer mediate the initial phase of wound repair, papillary fibroblasts are recruited at a later stage and dermal papilla and APM fibroblasts seem not to contribute at all [9,11,45]. Intriguingly, besides depositing/remodelling ECM in the wound bed, myofibroblasts are able to acquire a dermal papilla or adipocyte fate in response to distinct signals promoting hair follicle and adipocyte regeneration [46,47,48]. After tissue repair, wound bed fibroblasts re-establish a quiescent state to maintain skin homeostasis. The signals controlling distinct fibroblast lineage recruitment and promoting myofibroblast conversion to other mesenchymal cell populations are largely unclear. 

## 5. Skin Cancer

Skin cancers can be divided into cutaneous melanomas and nonmelanoma skin cancers (NMSCs). NMSCs are by far the most frequently diagnosed human cancers worldwide [49]. NMSCs comprise various types of carcinomas, such as basal cell carcinomas (BCCs), cutaneous squamous cell carcinomas (cSCCs), keratoacanthomas, Merkel cell carcinomas, cutaneous lymphomas, angiosarcomas and various rare adnexal tumours. Approximately 80% of nonmelanoma skin cancers are BCCs and 20% are cSCCs [49,50]. Both types of NMSCs originate from the epidermis. Although BCC is a malignant cancer, it is generally only locally invasive and rarely metastasises [49]. UV radiation is the principal mutagen in BCC pathogenesis, and dysregulation of the SHH/PTCH1/SMO pathway is central to BCC development [49]. Overexpression of the Hedgehog pathway, either through deletion of PTCH1, mutational activation of SMO, or overexpression of GLI1 or GLI2 have been reported in human and mouse BCC [19,49,50]. Compared to BCC, cSCC is a more aggressive tumour that can form lethal metastases and is associated with mutation in RAS GTPases (HRAS and KRAS), cell cycle regulators such as TP53 and CDKN2A and Notch signalling receptors (NOTCH1, NOTCH2, and NOTCH3) [13,14,49,51]. cSCC can be induced in mice through multistage carcinogenesis models employing either UV irradiation or chemical carcinogens or forced expression of oncogenes targeted to the epidermis [14,15,51]. Interestingly, studies using transgenic mice with expression of oncogenic KRAS in different compartments of adult epidermis revealed that both HF and IFE SCs represent the cells of origin of mouse cSCC [14]. However, oncogenic targeting of HF SCs led to formation of more aggressive and less differentiated cSCCs with features of epithelial to mesenchymal transition (EMT) [14]. Like cSCC, murine BCC can also arise from multiple epidermal compartments [51]. 

Cutaneous melanoma originates from pigment-producing melanocytes [52]. The incidence of cutaneous melanoma is substantially lower than that of NMSC; however, approximately 75% of skin cancer deaths are due to metastatic melanoma [52]. Malignant transformation into melanoma follows a sequential genetic model that results in constitutive activation of oncogenic signal transduction [52,53]. Oncogenic driver mutations in melanoma involve BRAF (~50%), NRAS, KRAS and HRAS (~25%) and NF1 (~15%) [52,53]. Other frequent genetic alterations include activating *TERT* promoter mutations, found in 30–80% of melanomas [52,53].

Communication between tumour cells and their microenvironment, including the tumour stroma (the non-transformed tissue components associated with a tumour), plays an important role in the development and progression of skin cancer [3,54,55,56,57]. Besides endothelial and immune cells, a major component of the microenvironment is cancer-associated fibroblasts (CAFs), which play an important role in the evolution of solid tumours. CAFs seem to originate from different mesenchymal populations, ranging from normal fibroblasts and mesenchymal SCs to transdifferentiated epithelial and endothelial cells. In contrast to normal fibroblasts, CAFs either reside within the tumour margin or infiltrate the tumour mass and show increased proliferation, migration, ECM deposition and secretion of growth factors and other ECM modulators [58,59]. To date there have been few studies of how different fibroblast lineages contribute to tumour stroma formation, and whether the tumour stroma differs between different types of skin cancers. Interestingly, one study showed that fibroblasts of the reticular dermis are predisposed to differentiate into CAFs upon cSCC signals, assisting invasion and EMT [60].

## 6. The Hippo Signalling Pathway 

The Hippo pathway is a highly conserved signal transduction pathway that regulates gene expression. The core of the pathway is a kinase cascade that in mammals comprises MST1 (Ste20-like kinase 1; also known as STK4) and MST2 (also known as STK3), the homologues of the *D. melanogaster* Hpo kinase, large tumour suppressor kinase 1 (LATS1) and LATS2 (Warts in *D. melanogaster*), the adaptor proteins Salvador 1 (SAV1) (Sav in *D. melanogaster*), MOB1A and MOB1B (Mats in *D. melanogaster*) and the paralogous transcriptional coactivator proteins YAP and TAZ (Yorkie in *D. melanogaster*) [61,62,63,64]. In addition to LATS1/LATS2, NDR1 (STK38) and NDR2 (STK38L) also function as YAP/TAZ kinases [65]. The predominant transcriptional binding partners of YAP/TAZ are TFs of the TEA domain (TEAD) family (TEAD1–TEAD4) (scalloped in *D. melanogaster*) [61,62,63,64].

The functions of YAP and TAZ appear to be largely but not entirely redundant [66]. Indeed, there is genetic evidence that—at least in certain tissue contexts—both paralogues might drive distinct transcriptomes [67,68]. Inactivation of YAP usually has stronger consequences on cellular physiology than of TAZ, but this might simply reflect different expression levels of YAP and TAZ [67]. 

Mechanistically, YAP/TAZ binds to gene enhancer elements in complex with a TEAD TF, interacting with chromatin remodelling factors and modulating RNA polymerase II to drive or repress the expression of target genes, which prominently include cell cycle, cell migration and cell fate regulators [69,70,71,72,73,74,75,76]. Although TEAD factors are their predominant transcriptional interaction partners, YAP/TAZ have been shown to physically interact with other TFs such as p73, RUNX1/2/3 and TBX5 [77,78,79,80,81,82,83,84,85]. YAP–TEAD complexes alone may likely not be sufficient to execute the different transcriptional programmes. Indeed, bioinformatics analyses of the regulatory regions that are bound by YAP/TAZ-TEAD complexes frequently identified cooperation between YAP/TEAD and other TFs [73,74,85,86]. Therefore, YAP/TAZ cooperates with various TFs and chromatin regulators to regulate target gene expression.

Upstream of YAP/TAZ, activation of MST1/MST2 induces the phosphorylation of SAV1 and MOB1A/MOB1B, which assists MST1/MST2 in the recruitment, phosphorylation and activation of LATS1/LATS2 [61,62,64]. In parallel to MST1/MST2, two groups of MAP4Ks (mitogen-activated protein kinase kinase kinase kinase), MAP4K1/2/3/5 (homologs of Drosophila Happyhour (Hppy)) and MAP4K4/6/7 (homologs of Drosophila Misshapen (Msn)), can also directly phosphorylate and activate LATS1/LATS2 [87,88,89]. Subsequently, LATS1/LATS2 phosphorylates YAP and TAZ on several serine residues. Of these sites, the most relevant residues that keep YAP/TAZ inhibited are S127 and S381 in human YAP and S89 and S311 in human TAZ [61,62,64]. YAP/TAZ S127/S89 phosphorylation by LATS1/LATS2 creates a binding site for 14-3-3 proteins which contribute to keeping YAP/TAZ in the cytoplasm [90,91,92]. However, in many cellular contexts this signalling input alone does not appear to be sufficient to inactivate YAP/TAZ, as several studies have documented S127/S89-phosphorylated YAP/TAZ in the nucleus [21,82,93,94]. S381/S311 phosphorylation of YAP/TAZ modulates their protein stability by triggering further phosphorylation by casein kinase 1δ or 1ε (CK1) as well as ubiquitylation by the SKP1-CUL1-F-box protein (SCF) E3 ubiquitin ligase complex and proteasomal degradation [90,91,92]. It should be noted here that Hippo pathway regulation is not static in either ON or OFF state, but rather it is dynamic. YAP/TAZ is under constant phosphorylation and dephosphorylation and is rapidly trafficked between the cytoplasm and the nucleus [95,96,97,98]. Altogether, these observations suggest that additional regulatory inputs need to work together with the core Hippo pathway to fully inhibit YAP/TAZ activity.

The activity of the Hippo pathway is regulated by a multitude of upstream inputs, many of which relay signals from the plasma membrane [62,63,64]. However, unlike other classical signal transduction pathways, such as the epidermal growth factor (EGF), transforming growth factor-β (TGFβ) or WNT signalling pathways, the Hippo pathway does not appear to have dedicated transmembrane receptors and extracellular ligands. Rather, the Hippo pathway is regulated by a network of upstream signalling components that have roles in other processes such as the establishment of cell adhesion [99,100,101,102,103,104], cell morphology [105,106,107,108] and cell polarity [109,110,111,112,113,114,115,116]. The activity of the Hippo pathway is thus modulated in response to mechanical strains and changes or defects in cell–cell and cell–ECM adhesion [63,64,117] but also nutrient availability and other cellular stresses [118]. Therefore, the Hippo pathway constitutes a sensor for tissue and cellular integrity rather than responding to dedicated extracellular signalling molecules.

The biophysical properties of the extracellular environment and the cell shape are profound regulators of Hippo pathway activity. Mechanical stress, such as that caused when cells are grown on stiff surfaces, triggers YAP/TAZ nuclear translocation, whereas detachment from the ECM causes YAP/TAZ nuclear export [93,102,105,106,107,119,120]. The effects of the mechanical properties of the ECM on the Hippo pathway are mediated by integrin complexes at cell–ECM adhesion sites (focal adhesions and hemidesmosomes) and changes in the actomyosin cytoskeleton induced by integrin signalling in response to physical ECM properties [100,102,103,121]. Although the mechanisms are not fully understood, mechanical forces may further regulate YAP/TAZ through modulating the structure of nuclear pores, and hence the nuclear translocation of YAP/TAZ [96].

The Hippo pathway is also modulated by extensive crosstalk with other signalling pathways. In particular, these include G protein-coupled receptors (GPCRs) that are activated by lipids (lysophosphatidic acid and sphingosine-1-phosphophate) or hormones (glucagon or adrenaline) and signal through F-actin to regulate YAP/TAZ [122,123]; the WNT pathway, which regulates YAP/TAZ through direct interaction with the β-catenin destruction complex and through destruction complex-independent mechanisms [124,125,126,127,128]; SRC family kinases that either promote YAP nuclear localisation and transcriptional activity directly by phosphorylating tyrosine residues or indirectly by repressing LATS1/LATS2 [94,100,121,129,130,131,132]; and the PI3K pathway [100,103,133,134]. 

## 7. Expression of YAP and TAZ in Skin during Development, Homeostasis, Regeneration and Cancer

In both mouse and human epidermis, there is a clear correlation between nuclear localisation of YAP and the extent of cycling activity of epidermal SCs. In mouse skin, YAP expression was demonstrated as early as embryonic day 12, when the epidermis exists as a single cell layer of undifferentiated progenitors (Figure 2B). At this time point, YAP was shown to be predominantly nuclear, suggesting it is already active as transcriptional coactivator [135]. On stratification, YAP remains mostly concentrated in the basal epidermal cell layer, but the numbers of basal cells with nuclear YAP localisation wanes as proliferative activity in the IFE gradually diminishes postnatally [100,135]. Similar changes in YAP expression patterns have also been documented in human epidermis [21,100]. The timing and patterning of TAZ expression during epidermal morphogenesis and in the postnatal epidermis appears to coincide with that of YAP [100]. During HF formation, YAP remains nuclear in the proliferative cell populations of the outer root sheet (ORS) and transit-amplifying matrix (Mx) cells, but is cytoplasmic in the terminally differentiated lineages of the inner root sheet (IRS) and hair shaft (HS) (Figure 2B) [135]. In adult epidermis, prominent yet cytoplasmic YAP expression can be detected in the lower HF in telogen when SCs are quiescent, but YAP becomes nuclear in the HF during growth phase (anagen) when SCs are actively cycling [136,137] (Figure 3). YAP is also expressed in basal cells of SGs, where its nuclear localisation correlates with the expression of the proliferation marker Ki67 [138]. In the adult IFE, YAP is cytoplasmic in suprabasal, differentiated cells and nuclear in basal layer cells, but there is ambiguity in the literature about the extent of nuclear YAP abundance in the basal layer [21,99,100,135,136,137,139,140]. It is worth noting that nuclear YAP abundance in the IFE and HF junctional zone increases as HFs enter anagen [137] (Figure 3), which is consistent with lineage tracing studies showing that basal IFE cells nearest to actively cycling HFs are considerably more proliferative than the ones more distant [141]. Given the variation in HF densities at different body sites, this may explain some of the differences in nuclear YAP abundance in adult mouse IFE reported by different laboratories. In skin fibroblasts YAP is nuclear in proliferative cells and is mainly cytoplasmatic in quiescent cells postnatally [21] (Figure 3). 

Upon skin wounding, increased numbers of cells with nuclear YAP/TAZ can be observed predominantly in the basal cell layer of the migrating epidermal tongue at the wound edge, but also in suprabasal cells [21,100]. This pattern is maintained after wound closure when the IFE is still hyperproliferative, but YAP (and supposedly also TAZ) becomes increasingly cytoplasmic again once tissue homeostasis has been re-established [21]. While YAP is mainly cytoplasmic in fibroblasts in adult dermis, it highly expressed and nuclear in dermal cells in and outside the wound bed in the early wound healing phase [21]. Increased epidermal YAP expression has also been documented in psoriasis, a chronic inflammatory skin disease inducing hyperproliferation and abnormal differentiation of epidermal keratinocytes [142]. Similarly, YAP is highly expressed and nuclear in fibrosis of the skin and other tissues, a disease condition characterised by excessive ECM deposition/remodelling, inflammation and skin cell proliferation [143]. 

YAP is also overexpressed in most types of human and murine epidermal cancers [101,135,139,144,145]. In cSCC, increased YAP expression was shown to correlate with disease progression [139,145]. Well-differentiated human cSCC presented nuclear YAP expression at the invasive tumour front, and poorly differentiated cSCCs with mesenchymal features showed a homogeneous stronger nuclear staining [139]. In contrast, TAZ was found to be expressed mainly in the cytoplasm of a subset of well-differentiated and poorly differentiated cSCC, and few cells stained positive for nuclear TAZ [139]. YAP is also strongly expressed and nuclear in spindle cell carcinoma (spSCC), a morphologically distinct type of cSCC with pronounced mesenchymal features [146]. In BCC, YAP is highly expressed and nuclear in superficial, nodular and infiltrative forms, while TAZ is mostly expressed in the tumour-surrounding mesenchyme [139,144,147,148]. In kerathoacanthoma, increased nuclear YAP expression has been observed in subset of tumours with low expression on the adherens junction component α-catenin [101]. YAP is also overexpressed in trichilemmal carcinoma and pilomatrixoma, rare tumours of HFs [149,150]. 

In a recent study, YAP protein expression was found to be elevated in benign melanocytic nevi and primary cutaneous melanomas, but YAP was present at only very low levels in normal melanocytes in healthy human skin [151]. Interestingly, more melanocytic cells in nevi and in early stage cutaneous melanomas had higher nuclear YAP abundance compared to cells from metastatic disease [151]. This study also found YAP to be ubiquitously expressed across a panel of melanoma cell lines, while TAZ expression was observed in most but not all cell lines [151]. In contrast, a different study found the expression of TAZ to be upregulated in human melanoma cell lines [152].

## 8. YAP/TAZ Drive Proliferation of Epidermal SCs and Fibroblasts during Development and Tissue Regeneration

In line with the timing and patterning of YAP expression during epidermal development, conditional knockout of YAP in Krt14-expressing epidermal progenitors of mouse skin (*K14-Cre/YAP* knockout mice) resulted in severe epidermal hypoplasia caused by insufficient proliferation of SCs to sustain epidermal morphogenesis, particularly in skin areas with high growth demand [99]. Vice versa, human Krt14 promoter-driven expression during mouse embryogenesis of a mutant YAP transgene (YAP-S127A, hereafter referred to as *K14*/*YAP-S127A*) with enhanced nuclear localisation led to the formation of a hyperthickened epidermis and impaired differentiation and invagination of HFs as a consequence of increased SC proliferation at the expense of terminal differentiation [99,135]. Accordingly, a genome-wide RNA interference (RNAi) screen identified YAP as an essential regulator of proliferation of human IFE SCs in culture, when they are actively cycling [21]. Expression of the YAP-S127A or NLS-YAP-5SA transgene (bovine Krt5 promoter-driven transgene with all five serine phosphorylation sites mutated and its N-terminus fused to a NLS; hereafter referred to as *K5*/*NLS-YAP-5SA*) both caused severe tissue dysplasia in adult epidermis that eventually progressed to tumour-like masses resembling cSCCs [99,146]. In stark contrast, adult mice expressing another YAP transgene (YAP-5SA-ΔC, lacking the transactivation domain and expressed under control of the bovine Krt5 promoter; hereafter referred to as *K5*/*YAP-5SA-ΔC*) displayed enhanced nuclear localisation and developed only a mild skin phenotype with later onset of epidermal hyperthickening and hair loss [136,137]. In fibroblasts, the YAP-5SA-ΔC transgene was found to display reduced transcriptional coactivator activity due to decreased nuclear accumulation [97]. However, YAP-5SA-ΔC showed strong nuclear accumulation in epidermal cells [137]. This suggests that sequences other than the C-terminus are involved in controlling nucleocytoplasmic shuttling in the context of keratinocytes. Interestingly, while *K14*/*YAP-S127A* transgenic mice displayed loss of terminally differentiated cell types in the IFE [99,135], the hyperthickening of *K5*/*YAP-5SA-ΔC* IFE was caused by expansion of both the basal and suprabasal cell compartments as well as hyperkeratinisation in the most differentiated cell layers [136]. This suggests that the C-terminus of YAP (including the YAP transactivation domain and PDZ-binding motif) may control the balance between epidermal SC proliferation and differentiation in the IFE. Consistent with the predominant nuclear localisation of YAP in SC-containing compartments during HF growth, *K5*/*YAP-5SA-ΔC* transgenic mice displayed striking HF abnormalities due to marked expansion of the SC populations in the lower HF [136]. In line with this, two weeks after tamoxifen-induced epidermal depletion of YAP and TAZ (*K5-CreERT*/YAP/TAZ), mice developed progressive hair loss and HF growth was completely blocked in neonates [100]. Apart from a moderate decrease in proliferation the IFE of adult *K5-CreERT*/*YAP*/*TAZ* double knockout mice showed no obvious abnormalities consistent with the lower nuclear abundance of YAP (and TAZ) in the basal cell layer of adult compared to foetal and neonatal mice [100]. Surprisingly, two other studies reported no obvious skin phenotypes in epidermis-restricted conditional YAP/TAZ double knockout mice [73,139]. This discrepancy can likely be explained by the different promoters used to drive conditional Cre transgene expression (bovine Krt5 promoter [100] vs. human Krt14 promoter [73,139]), which have different deletion efficiencies and onsets/timings [153,154,155]. Skin grafting experiments revealed that YAP knockdown significantly impaired SG development, and SGs were found to be grossly enlarged in *K5*/*YAP-5SA-ΔC* mice, pointing to a role of YAP in controlling SG homeostasis [136,138]. In contrast to the epidermis, the role of YAP/TAZ signalling in dermal fibroblasts during development and maturation remains largely unclear.

Consistent with the increased nuclear localisation of YAP/TAZ upon skin wounding, conditional YAP/TAZ knockout in the adult epidermis or topical application of interfering RNAs onto skin wounds slowed down wound closure due to reduced cell proliferation [100,156]. Similarly, RNAi-mediated YAP knock down in human primary keratinocyte cultures caused impaired regeneration of epidermal tissue in 3D organotypic skin cultures. The hypoplastic epidermis reconstituted by YAP knockdown keratinocytes also displayed premature onset of terminal differentiation, again highlighting the dual role of YAP in balancing SC proliferation and differentiation [21]. Interestingly, nuclear YAP abundance is prominent in basal cells throughout the wound healing zones of the regenerating epidermis [21], including the leading edge closest to the wound where cells are not proliferating but migrate as a sheet [4]. This suggests a role of YAP/TAZ in positively regulating keratinocyte migration, similar to what has been observed in other cell types [85,157,158]. In diabetic wounds with delayed healing, YAP expression is reduced, which can be recapitulated in vitro when dermal fibroblasts are cultured under high glucose condition [159]. In fibroblasts, YAP/TAZ knockdown attenuates key fibroblast functions, including matrix synthesis, contraction, proliferation on stiff matrix, whereas overexpression of activated mutants promotes fibroblast growth on soft matrix and drive fibrosis in vivo [143].

Interestingly, knock-in mice expressing a mutated version of YAP (YAP-S79A) that is unable to interact with TEAD TFs phenocopied the severe skin hypoplasia of YAP conditional knockout mice [99]. Likewise, RNAi-mediated silencing of TEAD expression was shown to significantly impair proliferation of primary mouse and human keratinocytes in culture [21,135]. These findings provided genetic evidence that TEADs are likely to be major transcriptional partners in epidermal SC proliferation. However, it cannot be ruled out that YAP/TAZ might interact also with other TFs and coactivators involved in controlling epidermal SC functions, as there is evidence that YAP/TAZ form a nuclear complex with SMAD2/3 [160] and mediate nuclear accumulation of activated β-catenin [137] in a human keratinocyte cell line. Indeed, the hypoplastic abnormalities in the epidermis of *K5*/*YAP-5SA-ΔC* transgenic mice were shown to involve activation of WNT16/β-catenin and TGF-β signalling [137,161,162] consistent with the well-established role of WNT and TGF-β signalling in controlling epidermal SC functions [163,164,165,166,167]. Although YAP is by default a transcriptional coactivator, it can also act as a corepressor in a complex with TEAD transcription factors and distinct chromatin-modifying proteins [76]. Thus, the increased expression of terminal differentiation markers observed in YAP knock down keratinocytes in culture [21,135] and in YAP/TAZ double knock out epidermis in vivo [119] could be directly linked to putative corepressor functions of YAP. YAP’s coactivator functions to drive TEAD-mediated gene transcription depend on its cofactor WBP2 [21]. While WBP2 knockout mice did not have hair growth abnormalities, they displayed reduced proliferation in the regenerating epidermis after skin wounding, thereby phenocopying some aspects of epidermis-specific loss of YAP/TAZ [21].

RNA sequencing followed by gene set enrichment analysis of human keratinocyte cell lines transfected with YAP-5SA or YAP-specific siRNAs led to identification of YAP-regulated gene sets that included cell cycle reactomes (such as E2F targets or cyclin E-associated genes) and cell growth reactomes (including transcriptional cell cycle regulators such as Myc, and global translation regulators) [100], similar to what has been found in other cell types [73,74]. Interestingly, the epidermal transcriptome of *K5*/*YAP-5SA-ΔC* transgenic mice was found to be enriched in genes that drive growth of cSCC cells [21,161]; consistent with the idea that aberrant YAP activation induces a premalignant tissue state that can progress further to neoplasia [99,136]. Among the key direct transcriptional targets of YAP/TAZ identified in the epidermis context are *Cyr61* (*CCN1*) [21,100,135], *CTGF* (*CCN2*) [21,119], *ITGB1* [100], *TGFBR3* [161] and the Notch ligands *DLL1* and *JAG2* [119]. Many of these YAP/TAZ target genes appear to function in positive feedback loops that maintain epidermal SC identity. For example, by controlling the expression of Notch ligands, YAP/TAZ appear to keep Notch receptors on the surface of epidermal SCs inactive (a process called cis-inhibition) and unable to receive differentiation-inducing Notch signals coming from neighbouring cells in-trans [119]. YAP/TAZ were previously shown to mediate alternative WNT signalling through the expression of secreted WNT inhibitors to suppress canonical WNT/β-catenin signalling [128]. However, in the epidermis of *K5*/*YAP-5SA-ΔC* transgenic mice, only WNT16 is induced by YAP, and it behaves as a canonical WNT ligand that activates β-catenin [162]. During skin wound healing, YAP/TAZ modulates the expression of TGF-β1 signalling pathway components, likely by acting in concert with AP-1 TFs and SMAD7 [156,168].

## 9. YAP/TAZ Activity in Skin is Controlled by Hippo Signalling-Dependent and Independent Signalling Mechanisms

All components of the core Hippo kinase cascade are expressed in both mouse and human keratinocytes [21,99,121,149]. However, there is ambiguity as to what extent Hippo signalling is involved in controlling the activity of YAP/TAZ in the epidermis. Conditional knockout of MST1/MST2 in mouse epidermis proved to be inconsequential for epidermal homeostasis and YAP activity [99]. In contrast, epidermis-restricted conditional knockout of MOB1A and MOB1B in postnatal skin led to gross abnormalities of IFE and HFs caused by marked expansion of the SC populations [149], reminiscent of the skin phenotypes of *K14*/*YAP-S127A* and *K5*/*YAP-5SA-ΔC* transgenic mice [99,135,136]. Indeed, dramatic expansion of cells with nuclear YAP was reported in the epidermis of MOB1A/MOB1B double knockout mice, and primary keratinocytes failed to exclude YAP from the nucleus in response to high cell densities, consistent with a defective Hippo pathway [149]. The consequences of conditional LATS1/LATS2 double knockout in the epidermis have not yet been studied. However, reduced expression and activating phosphorylation of LATS1/LATS2 in MOB1A/MOB1B keratinocytes in response to activation of upstream kinases [149], as well as increased YAP transcriptional activity upon RNAi-mediated LATS1/LATS2 ablation of [121], point towards involvement of LATS1/LATS2 in controlling YAP/TAZ in mouse epidermis (Figure 4). Interestingly knockdown of LATS1/LATS2 in a human keratinocyte cell line impacted nuclear YAP transcriptional activity only in confluent but not in dense cultures, indicating that cell compaction and reduced adhesive area are major triggers for YAP/TAZ localisation [21,106]. 

Indeed, several studies have identified the adherens junction component αE-catenin as a cell density-dependent YAP regulator [99,101,121]. These functions of αE-catenin involve its signalling properties but appear to be independent of cadherin-mediated adhesion [99]. Genetic ablation of αE-catenin in murine epidermis (using *K14-Cre* mice) or more specifically in the HF bulge (using *GFAP-Cre* mice) led to a hyperproliferative phenotype that was caused by increased nuclear abundance of YAP in the basal and suprabasal epidermal cell layers [99,101,121]. Two mechanisms of αE-catenin-mediated regulation of YAP/TAZ have been identified. In one mechanism, αE-catenin was found to promote cytoplasmic YAP localisation and S127 phosphorylation directly by modulating its interaction with 14-3-3 and the PP2Ac phosphatase [99] (Figure 4). This mechanism appears to operate also in SGs, where complex formation between αE-catenin, 14-3-3 and YAP is negatively regulated by Caspase 3-mediated cleavage of αE-catenin [138]. In a second mechanism, αE-catenin was found to supress SRC family kinase (SFK)-mediated tyrosine phosphorylation of YAP which otherwise promotes YAP’s nuclear localisation and TEAD binding [121] (Figure 4). Interestingly, both αE-catenin-dependent mechanisms appear to operate independently of LATS1/LATS2. When human keratinocytes were seeded onto ECM-coated polydimethylsiloxane elastomer substrates that mimic the epidermal–dermal interface [20], flattened cells with nuclear YAP where found to cluster at the tips of the substrates, while compacted cells with cytoplasmic YAP populated the sides and troughs of the substrates [21]. This patterning was shown to be dependent on mechanical forces exerted at intercellular junctions and modulated by SFKs and Rho GTPase signalling in response to undulations in the epidermal–dermal interface [21,169]. These findings suggest an additional layer of complexity in the regulation of YAP/TAZ by adherens junctions, namely that their stability and the organization and contractility of the associated actin cytoskeleton control the extent of cell compaction and thereby nucleocytoplasmic shuttling of YAP/TAZ independent of LATS1/LATS2 [21,170]. 

Integrin and ECM expression not only provide epidermal SC markers, but they also regulate SC fate during homeostasis, tissue repair and cancer progression [171,172]. When human keratinocyte cultures were fractionated by differential adhesion to extracellular matrix, expression of YAP was found to be highest in the rapidly adhering, ITGB1^hi^, SC-enriched, fraction [21]. In line with this, inhibition of ITGB1 with function-blocking antibodies or RNAi, or inhibition of the downstream effectors SRC and FAK or PI3K profoundly impaired YAP/TAZ nuclear localisation in a human keratinocyte cell line [100]. In vivo, conditional deletion of SRC or FAK in the epidermis, or pharmacological inhibition of SFK activity, led to decreased YAP levels and nuclear localisation in basal keratinocytes in skin [100]. Mechanistically, ITGB1-mediated YAP-activation appears to depend on the integrity and proper organisation of the F-actin cytoskeleton [100,173,174], but less so on actomyosin contractility [21,100]. Interestingly, ITGB1-mediated cell-ECM adhesion is stimulated by CTGF, one of the universal gene targets of YAP/TAZ/TEADs, suggesting a positive feedback loop that maintains epidermal SC identity [175]. Beside ITGB1 the hemidesmosome-associated ITGB4 [176] was also shown to control YAP activity via direct SRC-mediated phosphorylation of YAP [121]. Interestingly, ITGB4-mediated YAP activation is negatively regulated by αE-catenin [121], highlighting cross-talk between cadherins/catenins in adherens junctions and basal integrins as an important mechanism that coordinates SC cycling and terminal differentiation commitment during homeostasis and tissue repair [177].

In fibroblasts, increased ECM stiffness mechano-activates YAP/TAZ, which induces expression of profibrotic mediators such as PAI-1 and ECM proteins that provide a feed-forward loop maintaining fibroblast activation and tissue fibrosis [143]. Using photodegradable hydrogels of tuneable stiffness it was shown that YAP acts as a mechanical rheostat in mesenchymal SCs and promotes specific cell fates in response to past matrix stiffness, suggesting that YAP is involved in the regulation of mechanical cell memory [178]. While it is largely unknown if Hippo signalling differs in different fibroblast subpopulations, it was shown that LATS2 repressed preadipocyte proliferation and promotes adipocyte differentiation by inhibiting Wnt signalling and promoting PPARy transcriptional activity in the nucleus [179]. 

Summarised, YAP/TAZ are essential drivers of proliferation of epidermal SCs and fibroblasts during skin development and repair. The activity of epidermal YAP/TAZ is controlled by signalling downstream of adherens junctions and integrins as well as the mechanical forces transduced and imposed by their associated actin cytoskeleton. These mechanisms appear to work in concert with Hippo signalling in a context-dependent manner (Figure 4). Consistent with their solitary nature, mechanical signals seem to act as the predominant signalling cues in the regulation of YAP/TAZ in fibroblasts. 

## 10. YAP and TAZ as Oncogenes in Skin Cancers

In accordance with their increased expression in epidermal cancers, YAP and TAZ were found to play key roles in the development of cSCC and BCC. In a mouse model of cSCC, where activation of oncogenic *Kras^G12D^* in combination with *Tp53* deletion in the hair follicle lineage (using *Lgr5-CreER*/*Kras^G12D^*/*Tp53KO* mice) results in a wide spectrum of cSCC ranging from well-differentiated cSCCs to tumours resembling spSCC, YAP/TAZ were demonstrated to be essential for tumour initiation [139]. Upon conditional deletion of YAP/TAZ in *Lgr5-CreER*/*Kras^G12D^*/*Tp53KO* mice, cSCC/spSCC formation was completely abrogated, due to the rapid apoptosis of the oncogene expressing cells [139]. Likewise, conditional deletion of YAP/TAZ in a mouse model of BCC (*K14CreER*/*SmoM2* mice), which develops invasive BCCs post-expression of mutant SMO, efficiently prevented tumour initiation [139]. A similar study found that conditional deletion of YAP alone significantly reduced the tumour burden of *K14CreER*/*SmoM2* mice, but did not completely abrogate BCC formation [144]. This suggests that in the context of BCC YAP is the dominant paralogue, but TAZ might provide a compensatory mechanism in YAP-deficient BCC clones. Supporting the critical role of YAP in BCC development, those tumours that did form in *K14CreER*/*SmoM2* mice with conditional YAP deletion represented SmoM2-expressing clones that had escaped Cre-mediated recombination [144]. Longitudinal tracking of the evolution of YAP-positive versus YAP-null clones further demonstrated that the YAP-null clones were initially outcompeted by YAP-positive clones, and were eventually depleted over time as the tumours progressed to an invasive phenotype [144], potentially due to increased apoptosis [139]. Interestingly, YAP appears to promote the survival of BCC cells independently of WNT and Hedgehog signalling, the major signalling pathways involved in BCC development [144]. However, activation of the Hedgehog effector GLI2 in the epidermis *K5*/*YAP-5SA-ΔC* was shown to involve β-catenin [148]. YAP was shown to promote BCC initiation and progression via direct interaction with TEAD transcription factors to drive JNK-Jun signalling both at the level of *c-Jun* gene transcription but also upstream of c-Jun by controlling JNK activation [144]. c-Jun is a component of the functionally diverse AP-1 transcription complex, and in several cell types, YAP/TAZ/TEAD and AP-1 where shown to form a complex that synergistically activates target genes directly involved in the cell cycle control of S-phase entry and mitosis [73,74,85]. Indeed, ChIP sequencing analysis revealed co-occupation of chromatin regions by TEADs and AP-1 TFs in BCC [144]. Moreover, in a chemical carcinogen-induced mouse model of epidermal tumorigenesis, YAP/TAZ was also shown to be essential for tumour development [73]. Thus, YAP/TAZ/TEADs and AP-1 complexes appear to interact at multiple levels to promote epidermal tumour initiation and progression. 

YAP/TAZ and the TEADs were also found to play key roles in tumour progression of cutaneous melanoma. Several studies found that RNAi-mediated silencing of YAP/TAZ in human melanoma cell lines reduced cell proliferation, survival and anchorage-independent growth in 2D culture, dermal invasion in 3D organotypic skin cultures, and in vivo lung metastasis following tail vein injection [152,180,181]. Importantly, inhibition of YAP/TAZ function in melanoma cells was shown to overcome resistance to BRAF inhibitors [182,183,184]. Mechanistically, the resistance to BRAF inhibitors appears to involve evasion of the antitumour immune response by YAP/TAZ-driven upregulation of the immune checkpoint ligand PD-L1 [185]. However, a recent study revealed that while targeting of YAP and TAZ provided potent anti-melanoma effects against various human melanoma cell lines as well as uncultured, therapy-naive melanoma cells grown as tumours in patient-derived xenograft (PDX) assays, these effects were not evident in all PDX melanomas and cell lines [151]. Recently, somatic hypermutation of YAP was detected in one patient’s melanoma [151]. These first ever described hyperactivating YAP mutations in a human cancer manifested as seven distinct missense point mutations that caused serine to alanine transpositions. Four of these serines are key residues that are phosphorylated by the central Hippo pathway kinases LATS1/LATS2 and NDR1/NDR2 [61,62,64,65]. Consequently, the hypermutant *YAP* allele was shown to code for a highly active YAP protein [151].

Deregulation of the regulatory mechanisms that operate in healthy skin to control YAP/TAZ leads to their hyperactivation in skin cancers: Conditional knockout of αE-catenin in the HF bulge (using *GFAP-Cre* mice) was found to cause development of early onset keratoacanthomas displaying increased nuclear abundance of YAP [101], and there is significant correlation between low αE-catenin abundance and nuclear YAP localisation in human keratoacanthomas and cSCC [99,101]. Interestingly, conditional MOB1A/MOB1B double knockout mice developed trichilemmal carcinomas, but no other epidermal cancer types [149]. 14-3-3σ knockout mice displayed epidermal hyperplasia and increased formation of papillomas and cSCCs in response to chemical carcinogenesis that strongly correlated with enhanced nuclear YAP/TAZ abundance in the basal and suprabasal epidermal cell layers [186,187]. 

Similar to epidermal cancers, SFKs are also important drivers of YAP/TAZ activity in melanoma cells; however, in this context SFKs were shown to regulate YAP/TAZ indirectly by repressing LATS1/LATS2 [129]. In BRAF inhibitor-resistant melanoma cells, inhibition of actin polymerisation and actomyosin contractility was found to suppress both YAP/TAZ activity and drug-resistance, thus highlighting an important role of the actin cytoskeleton in this context [183]. 

The cancer microenvironment is characterised by elevated mechanical force at the cell and tissue levels. CAFs play a key role in the aberrant mechano-signalling observed in many tumour types [188]. Consistent with its role as a mechano-sensor, YAP is highly active in CAFs and essential to promote CAF-induced matrix stiffness, cancer cell invasion and angiogenesis. It is believed that similar to fibrosis, YAP establishes feed-forward loop between matrix stiffness and fibroblast activation maintaining CAF phenotype long-term. Mechanistically, matrix stiffness promotes actomyosin-dependent regulation of Src-family kinases to activate YAP by inducing a switch from 14-3-3 protein to TEAD1 and TEAD4 binding. YAP dependent expression of cytoskeletal regulators including ANLA, DIAPH3 and MYL9 then promotes further matrix remodelling and stiffening and thus maintaining YAP activation [120]. Similar to YAP–TEAD signalling the MRTF-SRF signalling axis also responds to extracellular signals and mechanical stimuli. Transcriptional comparison revealed that for the CAF contractile and pro-invasive phenotype both signalling pathways are essential and interdependent on the DNA level, suggesting that MRF-SRF and YAP–TEAD pathway interact indirectly by modulating the cytoskeletal dynamics /contractility [189]. Intriguingly, BCC progression in *K14CreER*/*SmoM2* mice was shown to be accompanied by activation of RhoA/ROCK signalling, fibroblast activation and ECM remodelling [148]. Therefore, in this context epidermal YAP may also be activated indirectly in response to the increased dermal stiffness due to epidermal Hedgehog signalling activity.

Beside extracellular signals including cell–cell contact, mechanical stress or matrix stiffness, also the metabolic status regulates YAP and LATS activity. Cellular energy starvation induces YAP inactivation via the key energy sensor AMPK through direct phosphorylation at S94 and indirectly through LATS kinase activation [190]. Recently, connections between ECM biophysical properties and metabolic cross-talk between cancer cells and CAF have been elucidated [191]. It was shown that CAF derived-aspartate supports cancer cell proliferation while cancer cell derived glutamate regulates the redox state of CAFs to promote ECM remodelling. This metabolic reprogramming seems to be dependent on the matrix stiffness and to be coordinated by YAP/TAZ through transcriptional regulation of glutaminase and the aspartate/glutamate transporter SLC1A3. In addition to promoting cell proliferation and contractility a role of YAP/TAZ in suppressing cell senescence has been described where YAP directly controls expression of key enzymes involved in deoxynucleotide biosynthesis [192]. 

In summary, YAP/TAZ signalling seems to be largely dispensable for skin homeostasis in adulthood but is essential for key processes of tumour initiation and progression in both epidermal and dermal cells. 

## 11. Targeting YAP and TAZ for Skin Cancer Treatment

In the clinic, the biggest challenge for BCC, cSCC and cutaneous melanoma remains treatment of patients with advanced or metastatic disease [13,49,50,52,193,194]. For comprehensive reviews on current treatment options see [194,195,196]. The inconsequentiality of YAP/TAZ inactivation for normal tissue function and their absolute requirement for cancer development and progression in the same tissue makes YAP/TAZ very attractive for cancer therapy, highlighting the possibility that targeting YAP/TAZ may display a large therapeutic window [197,198,199]. Since all upstream regulators ultimately impact on YAP/TAZ nuclear availability and transcriptional responses, designing compounds able to interfere at these levels may represent a ‘universal’ anti-YAP/TAZ strategy. One approach that could have potential as a YAP-targeting therapy for cSCC is pharmacological inhibition of SFKs using the drug dasatinib. In orthotopic mouse xenograft models, dasatinib treatment was shown to cause prominent inhibition of tumour growth through interference with SFK-induced YAP nuclear translocation and activation [121]. Of note, topical dasatinib application was recently found to induce regression of murine cSCC with less inflammation, no ulceration and no mortality compared to treatment with 5-fluorouracil, one of the standard chemotherapies for cSCC [200]. SFK inhibition-induced suppression of YAP/TAZ activity might also be beneficial for the treatment of metastatic melanoma [129]. This could be of particular importance in BRAF inhibitor-resistant melanomas, where drug resistance is conferred through YAP/TAZ activation [182,183,184]. In this context, it worth highlighting that one study found that interfering with the mechano-transducing functions of YAP/TAZ by inhibition of actin remodelling could suppress BRAF inhibitor-resistance [183]. Another strategy to interfere with YAP/TAZ functions that is currently under intensive investigation is to target the TEAD TFs [201,202,203]. Targeting the YAP/TAZ-TEAD complex should directly diminish the potential side effects expected from targeting the upstream proteins of the pathway which are interconnected with other signalling networks. Current strategies can be categorized into two primary approaches. One is to block the protein–protein interaction between YAP/TAZ and TEADs [204,205,206,207]. The other is to target a lipid pocket at the core of the TEADs, occupied by a palmitoyl ligand that is essential for TEAD folding, stability and YAP/TAZ binding [207,208,209]. The advantages and liabilities of disrupting the YAP/TAZ-TEAD complex through these two distinct mechanisms have yet to be fully elucidated. As initiation and progression of BCC and cSCC appears to critically depend on TEADs [73,144], small molecules able to interfere with YAP/TAZ-TEAD interaction could hold great promise for therapeutic interventions inthese cancers. 

## 12. Summary and Outlook

The research thus far shows that in the epidermis YAP and TAZ promote SC activation and cycling for development and regeneration of HFs and SGs as well as IFE morphogenesis, but they appear to be largely dispensable for IFE homeostasis in adult mice. Because current evidence of YAP/TAZ activation as a cue for SC mobilisation and tissue regeneration comes from studies of mice and cultured cell lines, at the moment, it is not clear whether primary human cells and organs can respond to YAP/TAZ activation in the same way as cells from current murine models. It should be noted here that—at least in culture—mouse and human keratinocytes display distinct growth behaviours [210,211]. It remains thus to be tested if the ambiguous findings related to the involvement of LATS1/LATS2 and other Hippo pathway components in the control of epidermal homeostasis and cancer development reflect such cell-intrinsic differences between mice and humans. Also, the kinase(s) involved in activating MOB1A/MOB1B and LATS1/LATS2 in keratinocytes still remain to be identified. 

Beside cell–cell contact and the mechanical microenviroment, the cell metabolism has been recently also identified to play a key role in YAP/TAZ regulation, suggesting that other regulatory mechanisms will be discovered in the future.

The transcriptional programmes executed by YAP/TAZ to promote epidermal SC cycling during development and tissue regeneration are still not fully characterised, as comprehensive YAP/TAZ ChIP sequencing studies have not yet been performed. Likewise, we still know little about the genes and processes under control of YAP/TAZ that drive nonmelanoma skin cancer initiation and progression. Curative therapeutic approaches for BCC and cSCC with distant metastasis are still lacking. Therefore, there is an urgent need to identify and understand the signalling pathways controlling initiation and progression of these epidermal cancers. Their widespread activation in human skin cancers and their essential function as a signalling hub pinpoint YAP/TAZ as prime candidates for effective cancer treatments. Since YAP/TAZ function to suppress terminal differentiation, their inactivation could potentially lead to ‘normalisation’ of cancer cells by reverting them from a more malignant to a benign differentiated phenotype. To meet the high demands on transcriptional regulators set by cancer cells to fuel their uncontrolled proliferation, YAP/TAZ need to interact with chromatin regulators, transcriptional cofactors, and even the basal transcriptional machinery [72,73,212]. Thus, an in-depth characterisation of the YAP/TAZ-associated machinery that drives this ‘transcriptional addiction’ in skin cancer cells and enables YAP/TAZ to execute their various downstream transcriptional programmes will help to inform new therapeutic approaches. 

In dermal fibroblasts, YAP/TAZ signalling is essential for sensing the physical environment of the cell and influences cell proliferation, ECM deposition and remodelling. Whether YAP/TAZ signalling is differentially regulated in different fibroblast subpopulations is currently unknown. Since YAP signalling is influenced by many other signalling pathways (including WNT and TGF-β1) it is tempting to speculate that YAP/TAZ signalling regulation and activity differs in papillary fibroblasts, characterised by high WNT signalling activity, and in reticular fibroblasts showing a pronounced ECM and immune gene signature [8]. Indeed, reticular fibroblasts have been identified as main drivers for aberrant ECM deposition/remodelling during scar formation and fibrosis, which is associated with increased YAP/TAZ signalling to establish a pathogenic feed-forward loop between matrix stiffening, proliferation and activation. While progress has been made in identifying the signalling pathways that contribute to fibrosis and cancer, we are still lacking a clear understanding of the early pathogenic processes in the dermis inducing and maintaining aberrant fibroblast function. Thus, dissecting the complex YAP/TAZ signalling crosstalk within different fibroblast populations during tissue homeostasis, regeneration and disease will help develop novel treatment strategies targeting aberrant fibroblast behaviour in fibrosis and cancer.

## 13. Materials and Methods

Figure 4 was created using Servier Medical Art templates, which are licensed under a Creative Commons Attribution 3.0 Unported License. 

## Figures and Tables

**Figure 1 cells-08-00411-f001:**
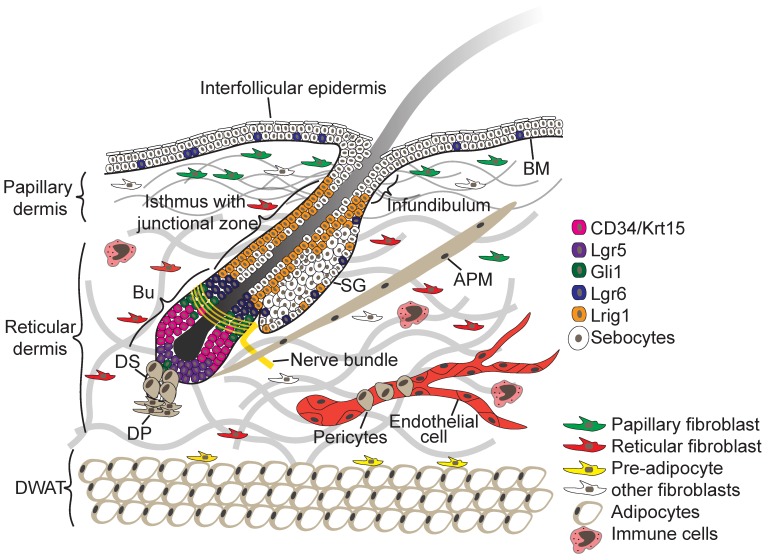
Morphology of the skin. The epidermis and dermis are separated by a BM. In the epidermis, multiple spatially distinct stem cell populations have been identified in the hair follicle bulge, isthmus and sebaceous gland, and their characteristic markers are shown in the colour coded legend. In the dermis, papillary fibroblasts are located in proximity to the BM and are embedded in thin collagen fibres. Reticular fibroblasts populate the central dermis and are surrounded by thick collagen bundles (grey). Preadipocytes are close to the DWAT where the mature adipocytes reside. In addition, specialised fibroblast subpopulations associate with the HF give rise to the DP, DS and APM. Endothelial cells form the blood vessels which are surrounded by pericytes. Sensory neurons are associated with the HF upper bulge SC population and different immune cell types populate different regions of the skin. Abbreviations: APM, arrector pili muscle; BM, basement membrane; Bu, bulge; DP, dermal papilla; DS, dermal sheath; DWAT, dermal white adipose tissue; SG, sebaceous gland.

**Figure 2 cells-08-00411-f002:**
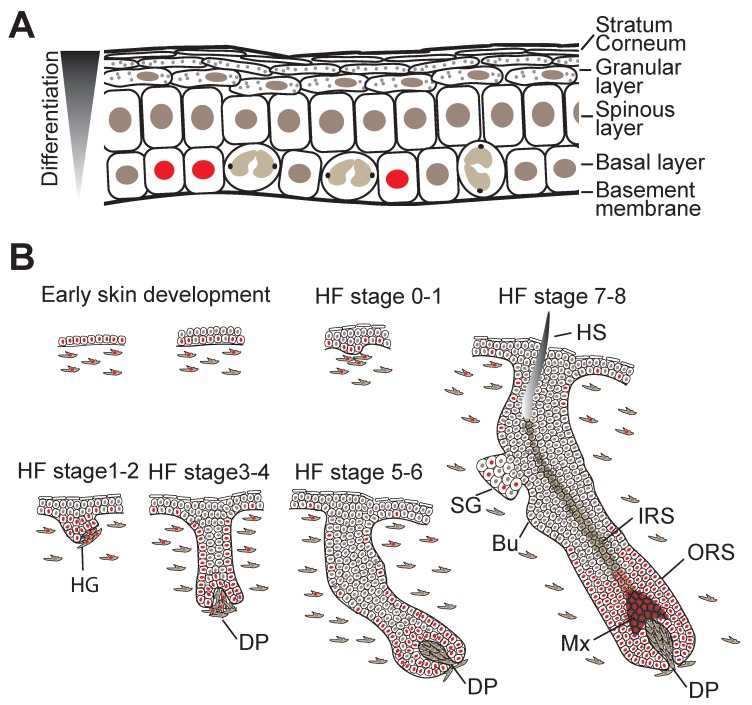
Yes-associated protein (YAP)/transcriptional coactivator with PDZ-binding motif (TAZ) activity in the IFE and HF development. (**A**) The IFE is a stratified squamous epithelium. It is divided into four main layers that are distinguished morphologically according to the differentiation status of the keratinocytes as they cease to proliferate and move upward to produce the skin’s barrier. Note that in adult IFE nuclear YAP/TAZ are restricted in cell clusters of the basal layer. (**B**) Early skin and HF development. During early skin development, cells of the epidermis and dermis are highly proliferative and highly positive for nuclear YAP/TAZ. Once the epidermis starts to stratify, only the proliferative cells in the basal layer maintain nuclear YAP/TAZ. HF development is initiated by an epidermal–mesenchymal cross-talk inducing condensation of mesenchymal cells beneath the BM which leads to the formation of a HF placode (HF stages 0–1). The HF placodes further matures into a hair germ (HF stages 1–2), which start to engulf the dermal papilla fibroblasts (HF stages 3–4). At HF stages 5–6 the dermal papilla is fully encapsulated; the HF epithelial cells differentiate into the distinct HF layers and the bulge and SG start to form. All cell compartments are clearly visible at HF stage 7–8 and the HF shaft emerges through the epidermis. YAP is nuclear in the placode and hair germ cells of the epidermis and dermis (HF stage 0–2) and then becomes more restricted to the highly proliferative basal epithelial cells (HF stage 3–8). YAP/TAZ are highly nuclear in the HF matrix, and there are additional cell cluster with nuclear YAP in the IFE and SG. Note that due to its wide-spread expression in skin, only nuclear YAP/TAZ are shown to indicate sites of YAPTAZ activity. Abbreviations: BM, basement membrane; Bu, bulge; DP, dermal papilla; HF, hair follicle; HG, hair germ; HS, hair shaft; IFE, interfollicular epidermis; IRS, inner root sheath; Mx matrix; ORS outer root sheath; SG, sebaceous gland.

**Figure 3 cells-08-00411-f003:**
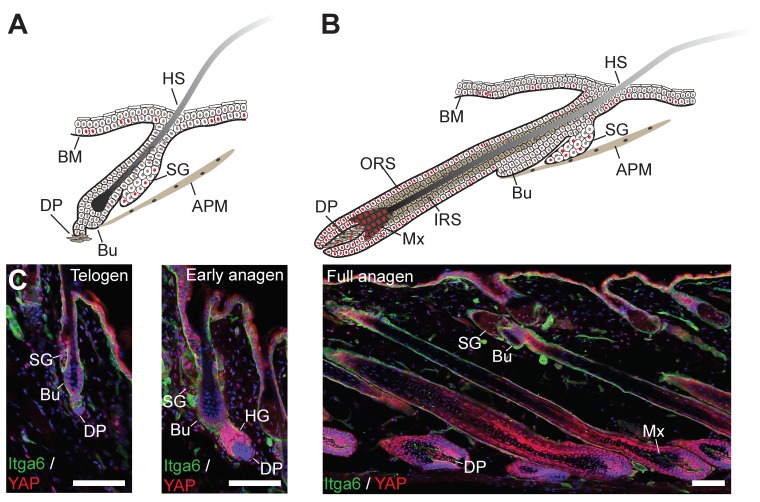
YAP localisation during HF cycling. (**A**,**B**) Schematic of a HF in the resting phase (telogen) (A) and growth phase (anagen) (B). In telogen phase, YAP is mainly localised to the cytoplasm in bulge and DP cells and there are only scattered cluster of cells with nuclear YAP in the IFE and SG. During anagen the highly proliferative cells of the ORS and HF matrix display strong nuclear YAP localisation, while YAP is cytoplasmatic in differentiating cells of the IRS and HS. Beside cell clusters with nuclear YAP in the IFE and SG during anagen, some DP cells also display YAP in the nucleus. Note that only nuclear YAP is shown. (**C**) Immunostaining for YAP (red) and Itga6 (green) of mouse skin during telogen, early anagen and full anagen. Nuclei are stained with DAPI (blue). Note, the strong increase in nuclear YAP in the HG and infundibulum during anagen induction. Scale bars, 50 µm. Abbreviations: Bu, bulge; DP, dermal papilla; HF, hair follicle; HG, hair germ; HS, hair shaft; IFE, interfollicular epidermis; IRS, inner root sheath; Mx matrix; ORS outer root sheath; SG, sebaceous gland.

**Figure 4 cells-08-00411-f004:**
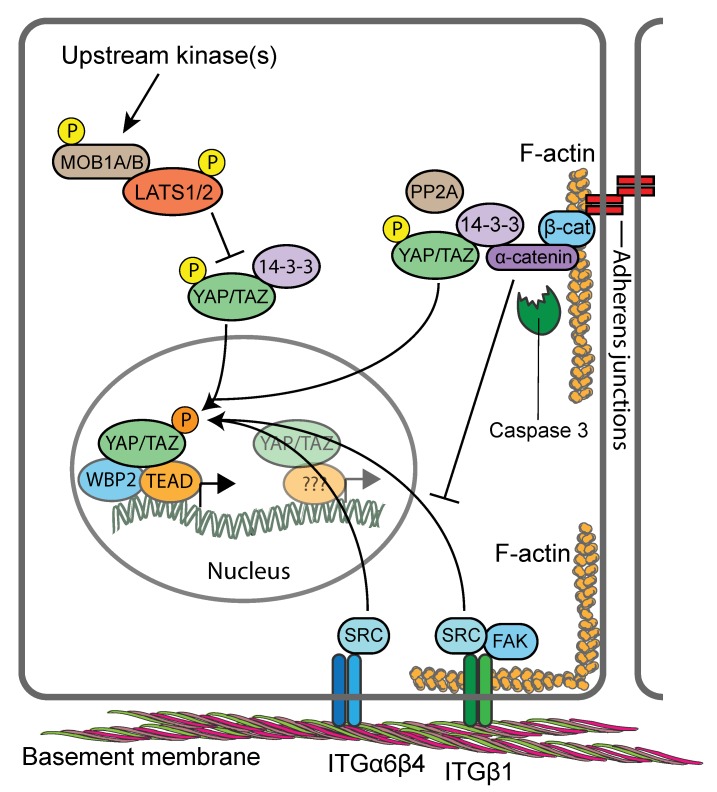
Regulation of YAP/TAZ in epidermal cells. Hippo signalling via MOB1A/MOB1B and LATS1/LATS2 inhibits YAP/TAZ via serine phosphorylation (yellow) to promote cytoplasmic retention. The kinases activating MOB1A/MOB1B and LATS1/LATS2 are not known. Integrin (ITG)–SRC signalling promotes YAP/TAZ nuclear localisation and TEAD binding. SRC can directly phosphorylate YAP/TAZ on tyrosine residues (orange) but may also act indirectly to activate Hippo signalling. A contractile F-actin-myosin cytoskeleton helps stabilise ITGβ1 adhesions and thus may contribute to SRC activation, while ITGβ4 adhesions are part of hemidesmosomal complexes that are stabilised by keratin intermediate filaments (not shown). At adherens junctions, α-catenin controls YAP/TAZ activity and phosphorylation by modulating its interaction with 14-3-3 and the PP2A phosphatase. In proliferating cells of the sebaceous gland, activation of caspase-3 cleaves α-catenin, thus facilitating the activation and nuclear translocation of YAP/TAZ. α-catenin can also inhibit ITGβ4-mediated direct activation of SRC. Putative nuclear interactions of YAP/TAZ with other transcription factors are also indicated.

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
