# Peer review of "The Roles of YAP/TAZ and the Hippo Pathway in Healthy and Diseased Skin"

_cells, 2019, doi:10.3390/cells8050411_

Round 1
Reviewer 1 Report
This is an excellent review article that nicely summarises recent progress in the field. I was impressed with the depth and breadth of thought, as well as with the useful synthesis of ideas. Overall, an outstanding achievement. I recommend publication without delay.
Author Response
We would like to thank the reviewer for his enthusiasm and kind words.
Reviewer 2 Report
In this manuscript the authors review the role of YAP/TAZ signalling in skin homeostasis, skin regeneration, and in skin disease. The first section gives a basic overview of skin biology. Then the authors review the role of YAP/TAZ signalling biology in the skin.
Many generic reviews on the role of YAP/TAZ in tissue regeneration and cancer development have been published in the past decade. But to my knowledge, this is the first focused review summarizing the research into the role of YAP/TAZ in skin biology, which makes it an important contribution that is of interest to the field.
I have two major critiques on this review:
1: The real focus of the review is described in manuscript sections 7-11, where sections 1-6 are merely basic background information that is extensively reviewed elsewhere, and frankly often much better. So please reduce these background sections to make this a more focused review on the role of YAP/TAZ in skin biology.
2: The referencing style is rather sloppy:
- The authors too often resort to citing other reviews rather than to primary research articles. In a revision I would like to see that predominantly primary research articles are cited, especially those published relatively recently in the skin (stem cell) biology and YAP/TAZ/Hippo field, to recognize the important contributions of the relevant authors.
- In a revision, please add citations in the relevant text sections of the following studies:
o Section 7: YAP expression in postnatal epidermis [76, 77, 79, 80].
o Section 7: YAP expression in NMSC: include Akladios et al., PloS 2017.
o Section 8: WNT16/beta-catenin signalling in response to epidermal YAP (Mendoza-Reinoso et al., SCR 2018).
o Section 10: YAP in NMSC: include observations described in [90], Akladios et al., PloS 2017.
- Many statements/findings in the manuscript are not backed by a citation. (lines 124-125, 128-131, 137-138, etc), or incorrect studies are cited ([78] in line 337, [80] in line 347, [94] in line 409, [76 in line 410]). Please ensure to correct these and others in a revision.
Section 8 does not review publications on fibroblasts as the title suggests. Please correct.
Author Response
The real focus of the review is described in manuscript sections 7-11, where sections 1-6 are merely basic background information that is extensively reviewed elsewhere, and frankly often much better. So please reduce these background sections to make this a more focused review on the role of YAP/TAZ in skin biology.
We thank the reviewer for the comments. Because this is the first focused review summarising the current knowledge about the Hippo pathway and the roles of YAP/TAZ in skin biology, we decided to a give a broader introduction. We believe this is helpful for readers in the Hippo signalling community who are not skin biologists. This clear separation will allow skin experts to immediately focus on the Hippo pathway-related parts of the review.
The referencing style is rather sloppy:
- The authors too often resort to citing other reviews rather than to primary research articles. In a revision I would like to see that predominantly primary research articles are cited, especially those published relatively recently in the skin (stem cell) biology and YAP/TAZ/Hippo field, to recognize the important contributions of the relevant authors.
We have made an effort to now cite mostly original articles throughout chapters 6 -12 (newly added references appear coloured red throughout the text). In the introductory chapters about skin biology, we feel it is more appropriate to refer our readers to the various excellent reviews written by our colleagues in the skin biology community.
- In a revision, please add citations in the relevant text sections of the following studies:
o Section 7: YAP expression in postnatal epidermis [76, 77, 79, 80].
o Section 7: YAP expression in NMSC: include Akladios et al., PloS 2017.
o Section 8: WNT16/beta-catenin signalling in response to epidermal YAP (Mendoza-Reinoso et al., SCR 2018).
o Section 10: YAP in NMSC: include observations described in [90], Akladios et al., PloS 2017.
The suggested original articles have now been cited in the respective chapters in our revised manuscript.
- Many statements/findings in the manuscript are not backed by a citation. (lines 124-125, 128-131, 137-138, etc), or incorrect studies are cited ([78] in line 337, [80] in line 347, [94] in line 409, [76 in line 410]). Please ensure to correct these and others in a revision.
We have added additional references throughout our revised manuscript where we found they were missing.
We have removed reference 78 in line 337 (now line 345).
We believe reference 80 in line 345 (now reference 137 in line 355) is correct, as the authors clearly show in their Figure 1 that nuclear YAP abundance increases in IFE, HF junctional zone and HFs during anagen similar to what we have also observed (Walko et al. Nat. Commun. 2017; and Figure 3 of our review).
We had included reference 94 (now reference 97), as it presents data showing that in fibroblasts, YAP-5SA-deltaC displays reduced transcriptional activity and nuclear accumulation, contrasting with the results from the Beverdam lab. We have re-phrased our statements in the revised manuscript (lines 424-426).
We believe reference 76 (now reference 135) is correctly supporting our statement. Figure 3 in Zhang et al. (PNAS, 2011) shows reduced thickness of loricrin-expressing cell layers as well as absent expression of HES1, indicative of impaired Notch-mediated spinous transition. We have also included Schlegelmilch et al. (Cell, 2011) as another reference supporting our statement in line 429.
Section 8 does not review publications on fibroblasts as the title suggests. Please correct.
We are well aware that there is a vast body of literature on YAP/TAZ signalling in fibroblasts of different tissues, which have been summarised in other reviews (e.g. Panciera et al. 2017; Noguchi et al 2018). To keep our review focused on skin, we decided to focus only on the latest findings related to skin fibroblasts, which are mainly restricted to wound healing. We have made this more clear now in the text of our revised manuscript.
Reviewer 3 Report
The author's described the physiology of the skin and how Hippo-YAP pathway play important functions during development and disease states. The review is overall well written with comprehensive and detailed information covering the field. Below are several suggestions that may improve the context and increase the impact of the current manuscript.
Related to Fig2 and relevant context, YAP is highly nuclear during early skin and HF development. It is also well established that Wnt is critical in these tissues. Both canonical and alternative Wnt can activate YAP. Subsequently, YAP secrets DKK1, Wnt5a to form negative feedback loop of b-catenin. It will be appropriate to discuss the possible role of Wnt-YAP in HF.
Authors have often cited multiple reviews rather than the original articles throughout the manuscript. For example, line 326, 3 review paper have been cited to explain Wnt, Src, PI3K in YAP regulation. Please cite the original articles that discovered canonical Wnt, alternative Wnt, Src, PI3K as YAP regulators.
YAP and TEAD are oncogenic in melanoma (EMBO J. 2016 Mar 1;35(5):462-78, Nat Genet. 2015 Mar;47(3):250-6, J Invest Dermatol. 2014 Jan;134(1):123-132). Add a paragraph to discuss the role of YAP in melanoma.
In Fig. 2 and 3, indicate which cell pictures are nuclear YAP or cytoplasmic YAP.
Please add a paragraph to discuss possible Hippo-targeting drugs that may benefit diseased skin.
Author Response
Related to Fig2 and relevant context, YAP is highly nuclear during early skin and HF development. It is also well established that Wnt is critical in these tissues. Both canonical and alternative Wnt can activate YAP. Subsequently, YAP secrets DKK1, Wnt5a to form negative feedback loop of b-catenin. It will be appropriate to discuss the possible role of Wnt-YAP in HF.
We thank the reviewer for his suggestion. We have included more discussion about the roles of WNT signalling in controlling YAP/TAZ in the epidermis in chapter 8, lines 500-503, in the revised manuscript.
Authors have often cited multiple reviews rather than the original articles throughout the manuscript. For example, line 326, 3 review paper have been cited to explain Wnt, Src, PI3K in YAP regulation. Please cite the original articles that discovered canonical Wnt, alternative Wnt, Src, PI3K as YAP regulators.
We have made an effort to now cite mostly original articles throughout chapters 6-12. In the introductory chapters about skin biology, we feel it is more appropriate to refer our readers to the various excellent reviews written by our colleagues in the skin biology community.
YAP and TEAD are oncogenic in melanoma (EMBO J. 2016 Mar 1;35(5):462-78, Nat Genet. 2015 Mar;47(3):250-6, J Invest Dermatol. 2014 Jan;134(1):123-132). Add a paragraph to discuss the role of YAP in melanoma.
We thank the reviewer for his suggestion. We have included a paragraph about cutaneous melanoma in chapters 5 and 7, lines 243-249 and 398-404, respectively, and paragraphs about the roles of YAP/TAZ in melanoma in chapter 10, lines 629-645 and 656-660, in the revised version of our manuscript.
In Fig. 2 and 3, indicate which cell pictures are nuclear YAP or cytoplasmic YAP.
YAP/TAZ are widely expressed throughout the skin. For simplicity, we are only showing nuclear YAP/TAZ in the illustrations of Figure 2 and 3 to indicate sites of active YAP/TAZ signalling. We are now stating this clearly in the respective figure legends. We have also slightly modified the colours in our illustrations to make nuclear YAP/TAZ localisation more visible.
Please add a paragraph to discuss possible Hippo-targeting drugs that may benefit diseased skin.
We thank the reviewer for his suggestion. We have included a new chapter (11) to discuss possible therapeutic strategies.
Round 2
Reviewer 2 Report
The authors have now improved their referencing style, but they decided to ignore my important other recommendation to reduce background sections 1-6.
These background sections have been topic of many other reviews (that the authors initially cited extensively themselves throughout the manuscript), but which were of much higher quality.
Since these background sections constitute over half of this manuscript, they actually distract from the real focus of this review.
This background information can easily be presented concisely in two paragraphs: one on skin biology, one on the Hippo/YAP/TAZ pathway. This would results in a better and more focused review.
Unfortunately, I cannot not recommend publication unless these background sections are summarized.
Reviewer 3 Report
The authors have made proper adjustments regarding the reviewer's comment and significantly improved the impact and quality of the manuscript.